# pH-dependent and dynamic interactions of cystatin C with heparan sulfate

Xiaoxiao Zhang [1,4], Xinyue Liu[2,4], Guowei Su[3], Miaomiao Li[1], Jian Liu[3], Chunyu Wang [2✉] & Ding Xu [1✉]

Cystatin C (Cst-3) is a potent inhibitor of cysteine proteases with diverse biological functions. As a secreted protein, the potential interaction between Cst-3 and extracellular matrix components has not been well studied. Here we investigated the interaction between Cst-3 and heparan sulfate (HS), a major component of extracellular matrix. We discovered that Cst-3 is a HS-binding protein only at acidic pH. By NMR and site-directed mutagenesis, we identified two HS binding regions in Cst-3: the highly dynamic N-terminal segment and a flexible region located between residue 70-94. The composition of the HS-binding site by two highly dynamic halves is unique in known HS-binding proteins. We further discovered that HS-binding severely impairs the inhibitory activity of Cst-3 towards papain, suggesting the interaction could actively regulate Cst-3 activity. Using murine bone tissues, we showed that Cst-3 interacts with bone matrix HS at low pH, again highlighting the physiological relevance of our discovery.

[1] Department of Oral Biology, The University at Buffalo, Buffalo, NY, USA. [2] Department of Biology, Center for Biotechnology and Interdisciplinary Studies, Rensselaer Polytechnic Institute, Troy, NY, USA. [3] Division of Chemical Biology and Natural Product, School of Pharmacy, The University of North Carolina, Chapel Hill, NC, USA. [4]These authors contributed equally: Xiaoxiao Zhang, Xinyue Liu. ✉email: wangc5@rpi.edu; dingxu@buffalo.edu

Cystatin superfamily members are reversible competitive inhibitors of papain-like cysteine proteinases. Cystatins share a common tertiary structure of an alpha helix that lays on top of an antiparallel beta sheet[1]. The cystatins can be categorized into family 1, 2, and 3, based on their molecular weight range. They play important roles in a variety of biological processes including tumorigenesis, stabilization of matrix metalloproteinases, and neurodegenerative diseases[2].

Cystatin C (Cst-3) belongs to family 2 cystatins, with a size between 13 and 14 kDa and two characteristic intrachain disulfide bonds. Encoded by a house keeping gene, Cst-3 is expressed by all nucleated cells and can be found in all tissues and body fluids, with particularly high concentrations in the cerebrospinal fluid[3,4]. The main function of cystatin C is to regulate the activity of cysteine proteases. Clinically, Cst-3 is used as a marker for evaluation of kidney disease because serum cystatin C concentration correlates with glomerular filtration rate[5]. Recently it has also been proposed as a predictor for cardiovascular risk[6].

Cst-3 is constitutively produced and secreted. In the endoplasmic reticulum, Cst-3 occurs as dimers, which are not active as a cysteine protease inhibitor. After they are released from the endoplasmic reticulum, they become monomeric Cst-3, which is functionally active as a cysteine protease inhibitor[7,8]. Monomeric Cst-3 inhibits cysteine proteases with a wedge-shaped structure, composed of N-terminal peptide Ser1–Val10, a central β-hairpin loop L1 and a C-terminal hairpin loop L2[9]. This specific wedge-structure is well-conserved and blocks the active site clefts of papain-like cysteine proteinases[10].

As a secreted protein, Cst-3 can be expected to perform some of its biological functions in the extracellular matrix. However, whether extracellular matrix components have any impact on the localization and activity of Cst-3 has never been investigated. Because both human and murine Cst-3 have a predicted pI > 8.5 and should carry multiple positive charges under neutral and acidic pH, we predict that Cst-3 might interact with negatively charged glycosaminoglycans in the extracellular matrix. Heparan sulfate (HS) is one of the most abundant and highly negatively charged glycosaminoglycans in the extracellular matrix[11]. It interacts with hundreds of proteins and modulates their activities[12]. Interestingly, histidine-rich glycoprotein protein, a cystatin superfamily member, has been shown to bind HS through its N-terminal cystatin-like domains, which suggests that the cystatin fold could bind HS[13]. To our knowledge, there has been no published study on the interaction between Cst-3 and HS or the structural details of this potential interaction.

In the current study we have demonstrated that Cst-3 is a bona fide HS-binding protein. Our data showed that Cst-3 is an unconventional HS-binding protein because it only binds HS under acidic conditions (pH ≤ 6.5). In addition, NMR spectroscopy and site-directed mutagenesis have revealed that the HS-binding site of Cst-3 is also unique, which comprises two most dynamic regions of Cst-3. Unexpectedly, we found the HS-binding interferes with the inhibitory activity of Cst-3 towards papain. Lastly, we provided histological evidence that Cst-3 indeed interacts with bone matrix HS in areas known to have acidic pH.

## Results

### Cst-3 binds heparin and HS in a pH-dependent manner.
To investigate whether Cst-3 is a HS-binding protein, we tried to express both human and murine Cst-3 in *E.coli*. While the majority of *E.coli* expressed murine Cst-3 existed in the monomeric form, all purified recombinant human Cst-3 existed in the dimeric form. As the dimeric form of Cst-3 is known to be inactive as a protease inhibitor[7,8], we chose to focus our study on

the monomeric form of murine Cst-3. We first examined the binding of recombinant Cst-3 to heparin using heparin–Sepharose chromatography. At pH of 7.1, Cst-3 did not bind heparin column and only appeared in the flow through. However, when we lowered the pH of running buffers to pH 6.5, 6 and 5.5, we observed progressively stronger binding to heparin Sepharose (Fig. 1a). To confirm this result, we also purified murine Cst-3 from overexpression in 293 cells and examined its binding to heparin Sepharose. Due to the low expression level of Cst-3 in 293 cells, binding of Cst-3 to heparin column was examined by western blotting. As shown in Fig. 1b (and Supplemental Fig. 1), the elution position of 293 cell expressed Cst-3 was identical to that of *E.coli* expressed Cst-3. Due to the much greater yield, *E.coli* expressed Cst-3 was used in all the following experiments.

As heparin is a highly sulfated form of HS, we next sought to determine whether Cst-3 actually binds moderately sulfated HS expressed by CHO-K1 cells using a flow-cytometry-based cell-surface binding assay. Interestingly, binding of Cst-3 to CHO-K1 cell surface was also pH-dependent, displaying no discernible binding at pH 7.1; in contrast, binding was observed at pH 6.5 (Fig. 1c). As expected, pretreatment of CHO-K1 cells with heparin lyase III completely abolished binding, suggesting that Cst-3 binding to CHO-K1 cell at pH 6.5 is strictly HS-dependent (Fig. 1d). Two mutant CHO cell lines with altered HS sulfation patterns were also tested for Cst-3 binding. Binding of Cst-3 to pgsE cells was reduced by 70% compared to its binding to CHO-K1 cells (Fig. 1e). As pgsE cells has ~50% reduction in overall sulfation[14], our result suggests that a minimal level of sulfation level of HS is required for Cst-3 binding. In contrast, the binding of Cst-3 to pgsF cells was comparable to its binding to CHO-K1 cells (Fig. 1e). As pgsF cells specifically lack 2-O-sulfation but maintains a similar level of overall sulfation as CHO-K1 cells[15], this result suggests that 2-O-sulfation is dispensable for Cst-3/HS interaction.

To examine heparin–Cst-3 interaction in greater detail, we performed surface plasmon resonance (SPR) analysis of the binding between Cst-3 and immobilized heparin at different pH. Our results showed that Cst-3 interacted with heparin at a $K_D$ of 0.6 μM and 1.8 μM at pH 5.5 and pH 6.5, respectively (Fig. 2a and b, Table 1). This result was consistent with our heparin Sephorase binding analysis, where we observed that higher salt concentration was required to elute Cst-3 from heparin column at pH 5.5 than at pH 6.5 (Fig. 1).

### HS oligosaccharide microarray analysis of Cst-3-HS interaction.
To understand the structural requirements of HS for interaction with Cst-3, we performed a HS oligosaccharide microarray analysis. The HS oligosaccharide array contains 52 structure-defined HS oligosaccharides with various lengths, sulfation levels and sulfation patterns (Fig. 3a and 3b). From the microarray analysis it is apparent that hexasaccharide is sufficient to bind Cst-3 (oligo #16, 34, and 40). Longer oligosaccharides, such as oligo #41, #20, and #27, did not show better binding to Cst-3 compared to hexasaccharide #34; therefore a hexasaccharide is sufficient to occupy the HS-binding sites of Cst-3. Interestingly, we found that the binding avidity between Cst-3 and HS did not strictly correlate with the overall sulfation level of HS, which is unusual among HS-binding proteins. For instance, heptasaccharides containing 4 and 5 sulfate groups (oligo #6 and #7, respectively) could already interact with Cst-3 quite well, while heptasaccharides containing 7 and 8 sulfate groups (#5 and #10, and #4 and #26, respectively), showed similar or even lower binding signals compared to oligo #6 and #7. For heptasaccharides, it appears that at least 4 sulfate groups are required for

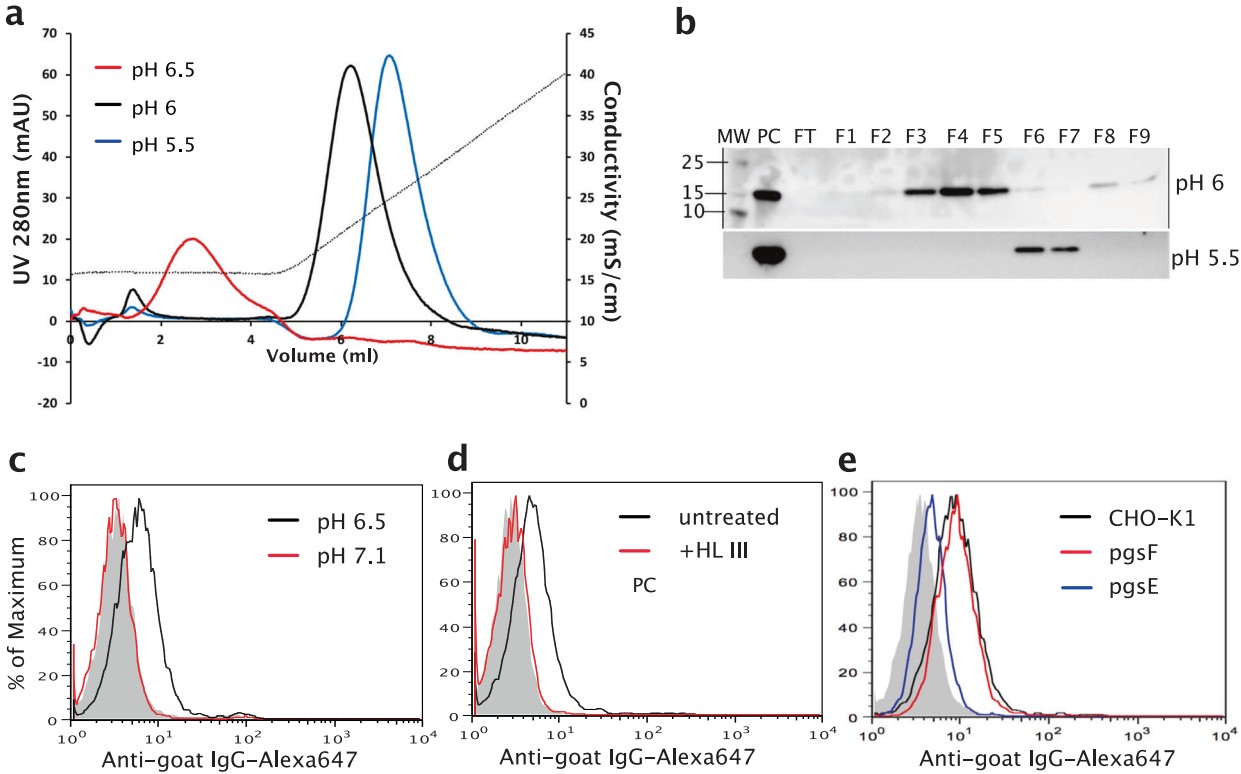

**Fig. 1 Cst-3 binds to heparin and HS in a pH-dependent manner. a** Binding of *E.Coli* expressed murine Cst-3 to heparin–Sepharose column at pH 5.5, 6, and 6.5. The dotted line represents the salt gradient (in conductivity mS/cm). **b** Western blot analysis of heparin–Sepharose chromatography fractions of 293 cells expressed murine Cst-3. The protein was loaded onto heparin–Sepharose column at pH 5.5 and pH 6 and 1 ml fractions were collected during the wash and elution. PC: positive control. FT: flow through. **c** Binding of Cst-3 (10 µg/ml) to CHO-K1 cells was determined by a FACS-based binding assay at pH 7.1 and pH 6.5. The bound Cst-3 were detected by staining with a goat anti murine Cst-3 antibody, followed by anti-goat-IgG–Alexa647. The shaded histogram is from cells stained only with primary and secondary antibodies. **d** Binding of Cst-3 to CHO-K1 cells pretreated with heparin lyase III (HL-III) at pH 6.5. **e** Binding of Cst-3 to pgsE and pgsF, two mutant CHO cells lines with altered HS structures.

binding to Cst-3, as heptasaccharides with three sulfate groups (#9, #12, and #53) showed minimal binding. At last, it appears that neither 6-O-sulfation nor 2-O-sulfation is strictly required for binding because oligos lacking either of these two modifications showed similar binding to Cst-3 (oligo #6 and #8).

**Structural analysis of Cystatin C–HS interaction by NMR.** To determine the potential HS-binding site of cystatin C, we performed $^1H,^{15}N$–HSQC NMR experiments by titrating HS hexasaccharide (oligo No. 34 on HS microarray, Fig. 3a) into Cst-3 at pH 5.5. From the HSQC spectra we observed HS-dependent chemical shift of a number of residues (Fig. 4a). After the HSQC peaks were assigned using 3D triple resonance experiments (Supplemental Fig. 2), we found that the largest chemical shift perturbation (CSP) was in the N-terminal region (K4 to E14) (Fig. 4b and 4c). In addition, a region between Q88 and C97 also saw substantial CSP upon titration with the hexasaccharide. To further confirm this result, we performed HSQC experiment by titrating cystatin C with heparin. Again, we found pronounced CSP in the N-terminal region and in the region between Q88 and C97 (supplemental Fig. 3). This result suggested that residues located in these two regions directly interacts with HS.

**Mutagenesis of the HS-binding site of Cst-3.** To determine the exact HS-binding site of Cst-3, we performed site-directed mutagenesis of a number of positively charged residues. Based on the NMR data, we first focused on basic residues located in the N-terminus (K4 and R8) and between Q88 and C97 (H90,

R93, and K94). Attempts to obtain K4A and R8A mutants were not successful as the mutations resulted in extremely poor solubility of the recombinant proteins. To solve this problem, we tried to express an N-terminus truncated form of Cst-3 instead (ΔN-Cst-3), lacking the N-terminal Ser1–Val10 decapeptide. When applied onto heparin column at pH 5.5, the ΔN-Cst-3 came out during column wash (150 mM NaCl, Table 2), suggesting that the N-terminal residues are indeed required for binding HS. Next, we examined whether H90, R93, and K94 are involved in binding to HS. Three additional basic residues, H86, R70, and K75, were also included in the mutagenesis study due their close spatial proximity to H90, R93, and K94. We found that H90A and R93Q-K94Q mutant were eluted from heparin column during column wash (Table 2), suggesting that these three residues likely directly participate in HS binding. In addition, R70A mutant also displayed 50 mM reduction in salt concentration required for elution, indicating that R70 too is involved in HS binding (Table 2). In contrast, the elution positions of K75A and H86A were almost identical to that of WT, which indicates that these two residues do not participate in binding to HS (Table 2). Of note, all identified HS-binding residues are conserved between murine and human Cst-3 (supplemental Fig. 4). To confirm that the mutations did not have a negative impact on the structural integrity of the Cst-3, we tested the activity of Cst-3 towards papain and found that with the exception of ΔN-Cst-3 mutant, the inhibitory activities of all other mutants were intact (Supplemental Fig. 5). As it is known that the N-terminal peptides directly participate in inhibition[16], we believe that the structural integrity of ΔN-

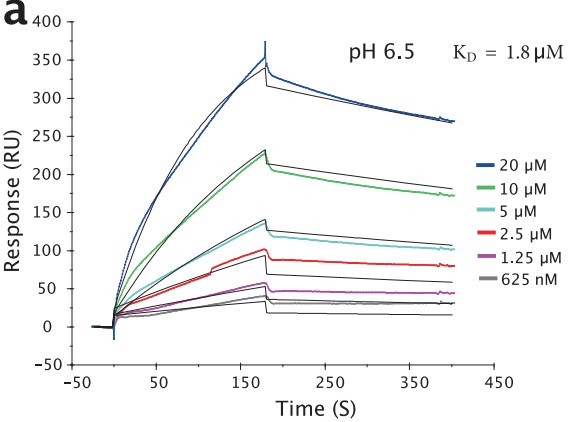

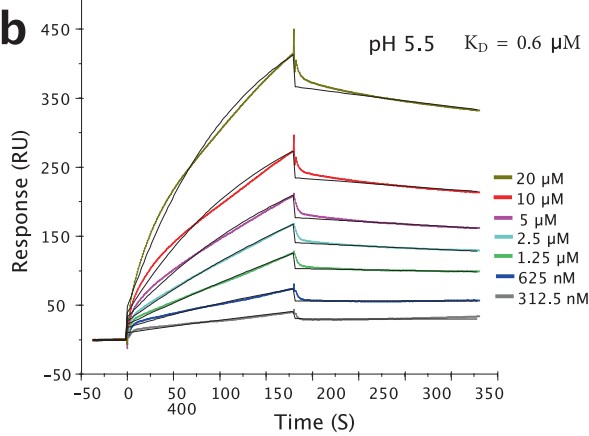

**Fig. 2 SPR analysis of heparin–Cst-3 interaction.** SPR sensorgrams of Cst-3 binding to immobilized heparin at pH 6.5 (**a**) and pH 5.5 (**b**). Notice although the SPR curve looked similar, the lowest concentration of Cst-3 used at pH 5.5 is half that of at pH 6.5 (312.5 nM versus 625 nM). The black curves are curves fit using BIAevaluate 4.0.1.

**Table 1 Summary of kinetic data of cystatin C–heparin interactions from SPR.**

| pH  | $k_a$ ($M^{-1}S^{-1}$) | $k_d$ ($S^{-1}$)       | $K_D$ (µM) |
|-----|------------------------|------------------------|------------|
| 6.5 | $4.3 \times 10^2$      | $7.6 \times 10^{-4}$   | 1.8        |
| 5.5 | $8.4 \times 10^2$      | $5.2 \times 10^{-4}$   | 0.6        |

Cst-3 was not compromised given the fact that it still maintained ~20% inhibitory activity compared to wild-type Cst-3 (Supplemental Fig. 5).

Next, we used the cell-surface binding assay to determine whether identified residues are truly involved in HS binding in a cellular setting. We found that the ΔN-Cst-3, H90A, R93Q-R94Q, and R70A mutants all showed substantial decrease in binding to CHO-K1 cell surface compared to that of WT Cst-3 (Fig. 5a–c). The results confirmed that these residues are indeed involved in Cst-3 binding to HS.

Further structural insights were gained by analyzing our data in the context of available crystal and NMR structures of monomeric Cst-3[17,18]. Our data suggest that R70, H90, R93, and K94, as well as N-terminal residues, participate in HS binding. In the crystal structure, R70, H90, R93, and K94 form a cluster of basic residues (Fig. 6a), which is typically found at the

HS-binding sites of most HS-binding proteins[12,19]. However, in all crystal structures of human Cst-3, the N-terminal residues [1]SSPGKPPRLVG[11] were missing, due to the highly dynamic nature of the segment. Indeed, in NMR structures of human Cst-3 published in 2020, it was found that the N-terminal segment is highly flexible and can adopt many different conformations (Fig. 6b)[17] Interestingly, several of these conformations could bring the N-terminus quite close to the H90 cluster of basic residues (Fig. 6a). Heparin and/or HS binding to Cst-3 likely select these conformations and shift the conformational equilibrium towards the formation of a complete HS-binding site, composed of two halves: the H90 basic cluster, and the N-terminal cluster, including basic residues K4 (K5 in human sequence) and R8, as well as the positively charged N-terminal amine. Based on this particular conformation of the N-terminal peptide, we calculated that the long axis of the proposed HS-binding site is 31 Å (between H90 and R8). As the typical length of HS hexasaccharide is around 27 Å, a hexasaccharide would just have sufficient length to occupy the proposed HS-binding site. We also expect a conformational change in Cst-3 upon HS binding, especially at and near the N-terminus, which could further shorten the HS-binding site in Cst-3 to fit the hexasaccharide.

The proposed HS-binding orientation revealed another basic residue, R51, might also participate in HS binding because its side chain directly faces the proposed HS-binding site (Fig. 6a). Indeed, R51A mutant also display greatly reduced binding to heparin column (Table 2), suggesting that it directly contributes to HS-binding.

**HS binding interferes with the inhibitory activity of Cst-3 towards cysteine protease.** Next, we tried to gain more functional insights on how HS/Cst-3 interaction might impact on the inhibitory activity of Cst-3 towards cysteine protease. To this end, we measured papain activity using a peptide substrate in the presence of apo Cst-3 or Cst-3/HS complex. Interestingly, we found that the heparin/HS-bound Cst-3 had greatly impaired inhibition of papain, displaying ~80% reduction in inhibition compared to apo Cst-3 (Fig. 7a). Intrigued by this observation, we repeated the experiment using heat-denatured BSA as an alternative substrate for papain. Again, we found that in the presence of HS hexasaccharide, Cst-3 displayed substantially reduced inhibition of papain. Combined, our data strongly suggest that under acidic pH, HS is capable of regulating the inhibitory capacity of Cst-3.

**Cst-3 binds extracellular HS in murine bone tissue.** To understand the physiological relevance of Cst-3/HS interactions, we examined the expression pattern of Cst-3 in murine femur sections at the knee joint by immunohistochemistry. Consistent with a previous report[20], Cst-3 is highly expressed by many types of bone cells including osteoclasts, osteoblasts, chondrocytes and a subset of bone marrow cells/leukocytes (Fig. 8a, dark brown staining; Fig. 8b shows negative control). In addition, Cst-3 staining was also found in the growth plate cartilage and newly formed bone matrix underneath the growth plate (diffuse brown staining). Interestingly, the extracellular staining of Cst-3 appears to be more intense around multinucleated osteoclasts, which is responsible for bone matrix resorption (Fig. 8a, red dotted line circled areas). As osteoclasts are capable of reducing the pH in the surrounding bone matrix well below pH 5 to facilitate decalcification and degradation of bone, we hypothesized that the bone matrix staining of Cst-3 in these areas could be due to binding of Cst-3 to extracellular HS. To test this hypothesis, we treated selected femur sections with heparin under pH 5.5 prior

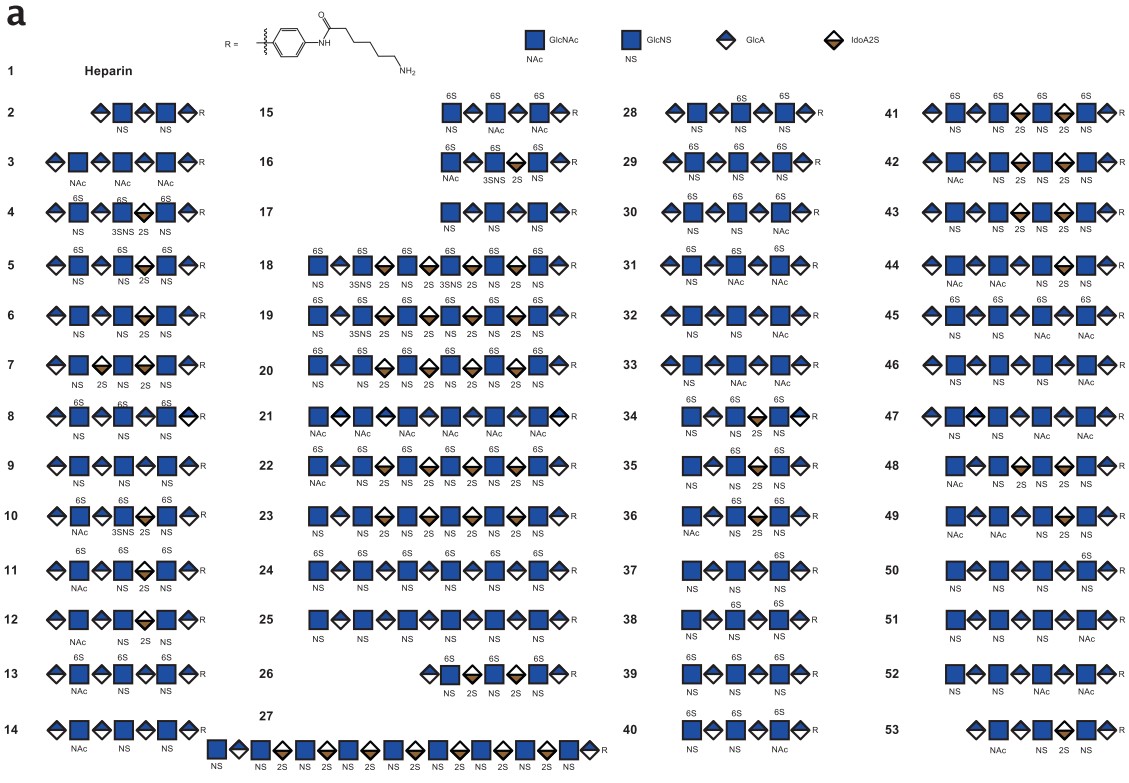

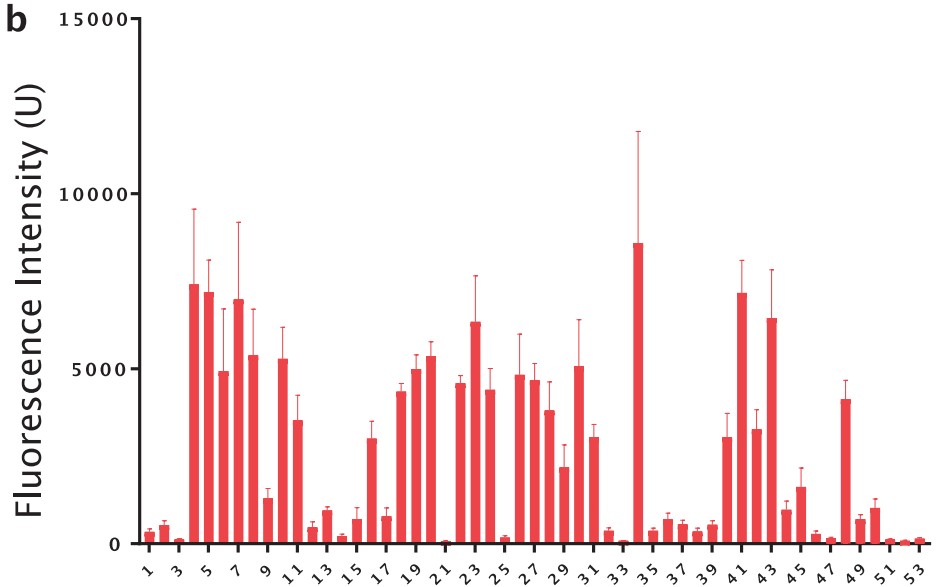

**Fig. 3 A hexasaccharide is sufficient to bind Cst-3 in HS oligosaccharide microarray. a** List of all 52 structure-defined oligosaccharides that are immobilized on the microarray chip. **b** Microarray was probed with 10 µg/ml Cst-3 at pH 5.5, and bound Cst-3 was stained with a biotinylated goat anti murine Cst-3 antibody followed by avidin conjugated with Alexaflour-488. The averaged fluorescence intensity was plotted. Error bar represents S.D. of 36 individual spots for each oligosaccharides. The source data of the microarray are included as Supplemental Data 1.

to staining with anti-Cst-3 antibody. Remarkably, we found that the bone matrix staining of Cst-3 was markedly reduced around the osteoclasts (Fig. 8c–d), which suggests that soluble heparin was able to compete with extracellular matrix HS for Cts-3 binding. In contrast, the cytoplasm staining of osteoclasts, chondrocytes and osteoblasts were largely unchanged after pre-treatment of heparin under acidic pH, suggesting heparin treatment did not impact Cst-3 immunostaining nonspecifically. Our data strongly suggest that Cst-3 is capable of binding to extracellular HS in the bone matrix, in particular at low pH regions

surrounding osteoclasts. This interaction could potentially regulate the physiological functions of Cst-3 in maintaining a fine balance between bone resorption and formation, such as inhibiting the activity of the cathepsin K, the major collagenase responsible for bone matrix degradation

### Discussion

In the current study we have shown that cystatin C, a ubiquitously expressed member of cystatin superfamily, is a HS-binding

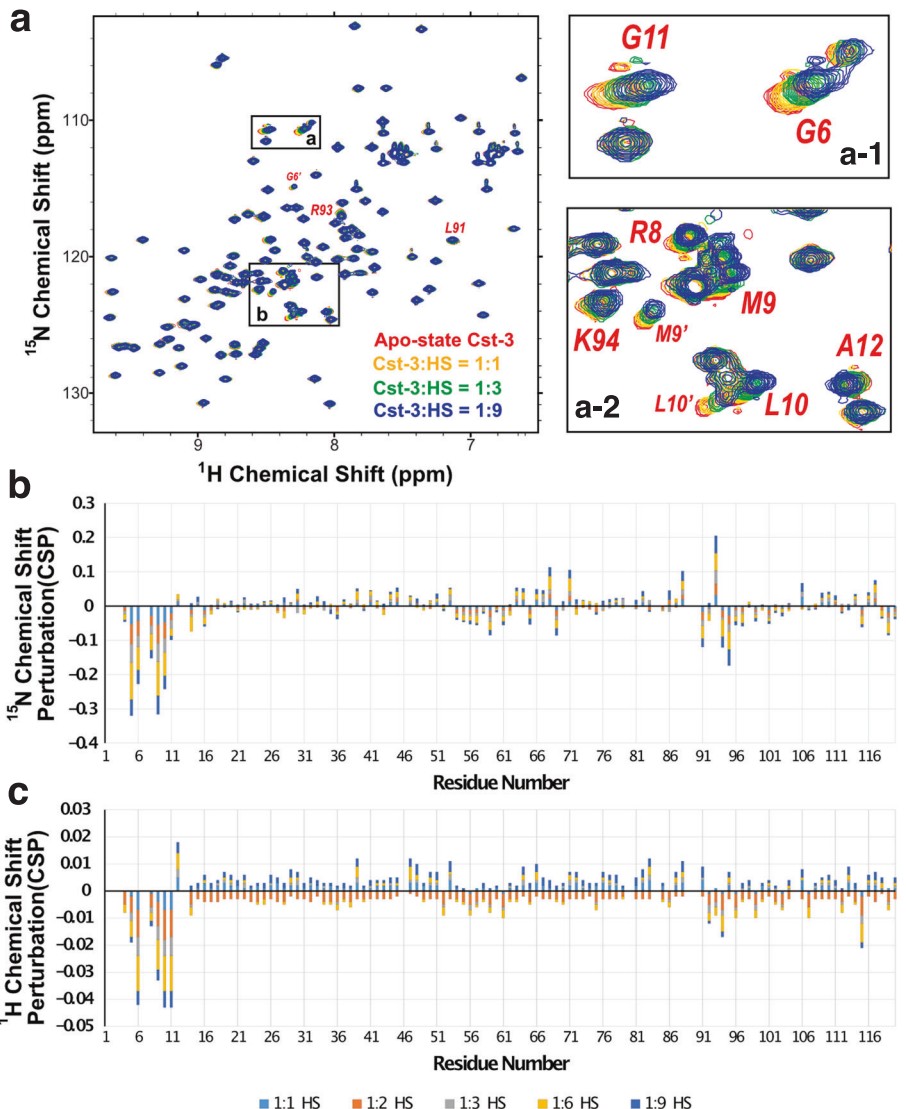

**Fig. 4 NMR titration analysis of Cst-3 and HS hexasaccharide identified two binding sites: the N-terminus and a region around H90. a** HSQC spectra of Cst-3 when it was titrated with 1:1 to 1:9 molar ratio of Cst-3: hexsaccharide at pH 5.5. **b** $^{15}$N-chemical shift perturbation plot of all residues. **c** $^{1}$H-chemical shift perturbation plot of all residues.

| Table 2 Comparison of WT and mutant cystatin C binding to heparin–Sepharose column at pH 5.5. | |
|---|---|
| **Protein** | **Elution NaCl concentration** |
| WT monomer | 280 mM |
| N truncated | 150 mM (during wash) |
| R70A | 230 mM |
| K75A | 270 mM |
| H86A | 270 mM |
| H90A | 150 mM (during wash) |
| R93Q-K94Q | 150 mM (during wash) |
| R51A | 240 mM |

protein. However, unlike conventional HS-binding proteins, which bind HS well at neutral pH, Cst-3 only binds HS at acidic pH. Using several biochemical and biophysical methods including SPR, heparin–Sepharose chromatography and cell-surface HS binding, we have shown that Cst-3 displayed progressively stronger binding to HS at lower pH from pH 7.1 to pH 5.5. This pH-dependence likely stems from protonation of a histidine at acidic pH, resulting in neutral to positive charge conversion. Indeed, our mutagenesis study has identified a critical histidine residue (H90) that is required for HS binding. It is important to note that Cst-3 is not the only HS-binding protein that displayed acidic pH-dependence. In fact, histidine-rich glycoprotein protein, which contains two cystatin-like domains, has been shown to bind HS in an acidic pH-dependent manner[21,22]. Other examples include selenoprotein P[23], serum amyloid A[24], diphtheria toxin[25], and gp64[26]. Interestingly, a critical role of histidine residues has been demonstrated in acid pH-dependent HS binding of histidine-rich glycoprotein protein, selenoprotein P and amyloid A, which strongly suggests that incorporating histidines into the HS-binding sites is likely a universal mechanism for pH-dependent HS binding.

Our mutagenesis study confirmed that the residues involved in HS binding segregate into two clusters (Fig. 6a). One cluster is located at the N-terminus, including K4, R8 and the N-terminal amine. The other cluster is located around H90, which includes

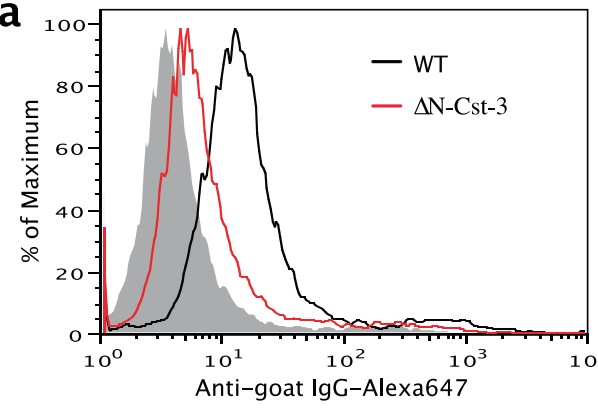

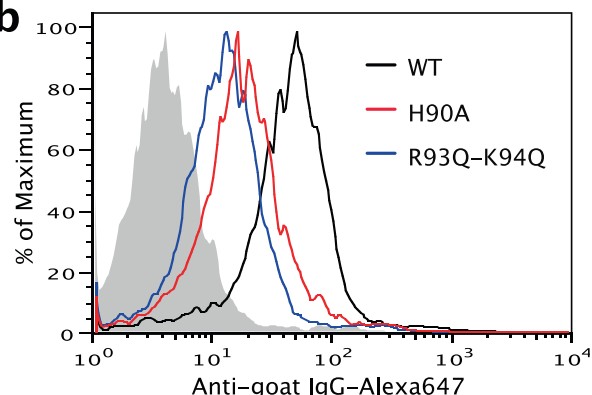

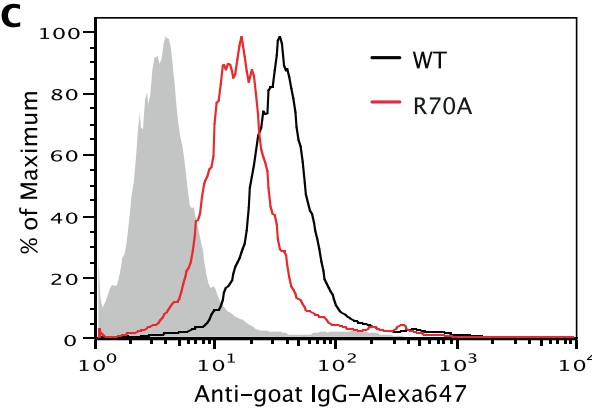

**Fig. 5 Binding of Cst-3 mutants to cell-surface HS.** Binding of ΔN-Cst-3 (**a**), H90A and R93Q-K94Q (**b**), and R70A (**c**) mutants to CHO-K1 cells were determined by FACS. All proteins were used at 30 μg/ml and bound Cst-3 were detected by a goat anti murine Cst-3 antibody followed by anti-goat-IgG–Alexa647.

R70, H90, R93, and K94. As both clusters are required for HS/heparin binding, we reasoned that the two clusters must come together to form one HS-binding site rather than function independently. Interestingly, a very recent NMR study of Cst-3 has found that the N-terminal region of Cst-3 is highly flexible, explaining why the structure of the N-terminus is undefined in multiple crystal structures[17,18] (Fig. 6b). The same study found that the region around H90 is also highly dynamic. In the NMR structure ensemble, the N-terminus can adopt conformations either pointing away or towards the H90 cluster (Fig. 6b)[17]. Of particular interest, the study has identified several NOE effects between the N-terminal segment and the residues around H90,

which suggest that the N-terminus could already weakly interact with the regions around H90 in the absence of HS. HS binding likely induces the N-terminus and H90 clusters to adopt conformations that are conductive for bringing together basic residues located at the N-terminus and the H90 cluster. In another words, HS binding can only occur when the N-terminus is in a conformation that allows proximity of the N-terminal segment to R70, H90, R93 and K94, which would form a complete HS-binding site of Cst-3. Such binding model could also explain the large chemical shift we observed at the N-terminal segment when Cst-3 was titrated with HS/heparin. The CSPs caused by HS or heparin binding at the N-terminus were negative in both $^{15}$N and $^1$H dimension, consistent with an increase in helical propensity upon HS binding[27]. To our knowledge, no other HS-binding protein has been shown to incorporate two highly dynamic regions into one HS-binding site. This knowledge further exemplifies the highly diverse binding modes that HS-binding proteins can adopt to interact with HS.

The requirement of the N-terminal peptide for HS binding could partially explain our observation that HS-bound Cst-3 displayed greatly reduced antipapain activity (Fig. 7). It was reported previously that the N-terminal peptide of Cst-3 is required for the inhibition of several papain family cysteine proteases[16]. Our analysis also confirmed that ΔN-Cst-3 mutant displayed severely impaired inhibition towards papain (Supplemental Fig. 5). As the N-terminal peptide is required for both papain inhibition and HS binding, it is likely that these two molecular interactions are mutually exclusive. Our study suggests that under acidic pH, HS could regulate the biological functions of Cst-3 as a cysteine protease inhibitor by inhibiting its interactions with cysteine proteases.

Acidic extracellular environment can occur under both physiological and pathological conditions. One physiological process that requires acidification of extracellular environment is bone resorption, a process essential for maintaining bone health. During bone resorption, osteoclasts lowers the local extracellular pH down to 3–6 to allow resorption of calcium hydroxyapatite[28,29], the main inorganic component of bone. To degrade collagen I, the main organic component of bone, osteoclasts secrete cathepsin K (a cysteine protease) into the resorption pit, whose collagenase activity is most robust around pH 5.5–6.5[30,31]. Because osteoclasts also secrete Cst-3 into the resorption pit and there is evidence that Cst-3 co-localizes with cathepsin K in certain areas within the resorption pit and surrounding bone matrix[20], Cst-3 likely plays a role in inhibiting the activity of cathepsin K during bone resorption. As Cst-3 interact with HS under this acidic environment (Fig. 8), and that cathepsin K is a known HS-binding protein[32], it is possible that HS might regulate the interaction between Cst-3 and cathepsin K during bone resorption. Further studies are needed to understand the detailed mechanisms of how HS regulates Cst-3 and cathepsin K interaction. Acidosis conditions have also been found during various pathological conditions. It was reported that infection and inflammation can reduce the local extracellular pH to as low as pH 5.5[33,34]. Also, ischemia could reduce the local tissue pH down to pH 6 to pH 6.5[35,36]. Finally, the microenvironment of most solid tumor is acidic due to poor diffusion and increased fermentative metabolism, and the extracellular pH as low as pH 6.5 has been reported[37]. As Cst-3 has been reported to play roles on all above-mentioned pathological processes[6,38–44], it is conceivable that HS might regulate Cst-3 functions in these processes.

In conclusion, we have shown that Cst-3 is an acidic pH-dependent HS-binding protein. Our structural studies revealed that Cst-3 adopt a unique HS-binding mode, where the binding site consists of two distant halves and involves the highly dynamic

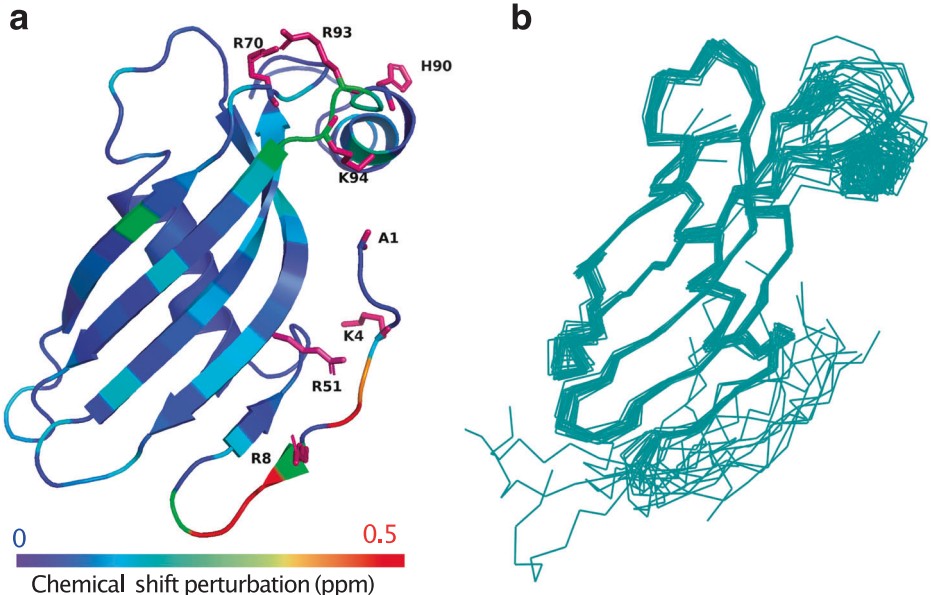

**Fig. 6 HS binds to positive charges in the two most flexible regions of Cst-3: the N-terminus and a region around H90. a** Cst-3 ribbon was colored based on NMR chemical shift perturbation (CSP) upon HS oligosaccharide titration. Positively charged residues in the N-terminus and near H90 are shown in pink stick model, including A1 (with charged N-terminus amino group), K4, R8, R51, R70, H90, R93, and K94. This structure was generated from homology modeling of model 2 of the NMR structure of human Cst-3 (PDB:6RPV). **b** NMR structure ensemble of the full-length monomeric human Cst-3 (PDB:6RPV), demonstrating the flexibility of the N-terminus and the region around H90.

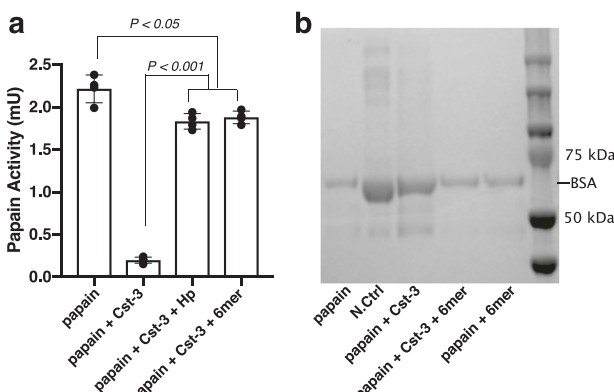

**Fig. 7 HS interferes with the biological function of Cst-3. a** The enzymatic activity of papain was determined by using a colorimetric peptide substrate in the presence or absence of Cst-3, and in the presence or absence of HS hexasaccharide (6mer) and heparin (Hp). $n = 4$. **b** Heat-denatured BSA was digested with papain in the presence or absence of Cst-3, and/or in the presence or absence of HS 6mer. N.Ctrl lane was run with only denatured BSA. Data are representative of at least three similar experiments.

N-terminus and a critical histidine residue H90 in a dynamic region. By identifying HS as a novel pH-dependent binding partner of Cst-3, our study brought a new perspective to the structural mechanisms of cystatins in both physiological and pathological processes.

## Methods

**Expression and purification of full-length mouse cystatin C in *E.coli*.** Complete open reading frame of murine or human Cst-3 was cloned into pET21b using NdeI and XhoI restriction sites. Recombinant Cst-3 was expressed in Origami-B cells (Novagen) with co-expression of *E.coli* chaperon proteins GroEL-GroES at 22 °C. Purification was carried out using HiTrap SP cation exchange column at pH 6.5

(MES buffer), followed by gel permeation chromatography on a Superdex 200 column (GE healthcare) in 20 mM Tris, 150 mM NaCl, pH 7.1. Superdex200 was able to provide baseline separation of monomeric and dimeric murine Cst-3 and only monomeric fraction of recombinant Cst-3 was used in this study. In contrast, all purified human Cst-3 existed in the dimeric inactive form. After purification, cystatin C was >99% pure as judged by silver staining.

**Heparin–sepharose chromatography.** To characterize the binding of WT Cst-3 and Cst-3 mutants to heparin, 100 μg of purified monomeric WT or mutant Cst-3 was applied to a 1 ml HiTrap heparin–Sepharose column (GE Healthcare) and eluted with a salt gradient from 150 mM to 1 M NaCl at pH 5.5 (NaOAc buffer), pH 6 and pH 6.5 (MES buffer), or pH 7.1 (HEPES buffer). The conductivity measurements at the peak of the elution were converted to the concentration of NaCl based on a standard curve.

**Expression and purification of Cst-3 in mammalian cells.** Complete open reading frame of murine Cst-3 was cloned into pcDNA3.1 (Invivogen) using XhoI and ApaI restriction sites. Recombinant Cst-3 was expressed in 293-Freestyle cells (Thermofisher) by transient expression using FectoPRO transfection reagent (Polyplus transfection). Secreted Cst-3 was analyzed by heparin–Sepharose chromatography as described above.

**Flow-cytometry-based binding assay.** CHO-K1 cells were incubated with WT or mutant Cst-3 in 100 μl of MES buffer containing 0.1% BSA for 1 h at 4 °C. After rinsing, bound Cst-3 was stained with goat anti-mouse cystatin C (5 μg/ml, AF1238, R&D systems) for 1 h at 4 °C, followed by anti-goat IgG–Alexa647 (1:500, ThermoFisher Scientific) for 30 min and analyzed by flow cytometry. In some experiments, cells were pretreated with recombinant heparin lyases III (10 milli-units/ml) for 15 min at room temperature prior to binding experiments.

**Surface plasma resonance.** Biotinylated heparin was immobilized to a streptavidin chip based on the manufacturer's protocol. The successful immobilization of HS was confirmed by the observation of a 100–200 resonance unit increase in the sensor chip. The control flow cell 1 was blocked by a 1-min injection with saturated biotin. Different dilutions of Cst-3 samples (concentrations from 0.3 to 20 μM) in NaOAc or MES buffer (pH 5.5 and pH 6.5, respectively) were injected at a flow rate of 30 μl/min. This is followed by a 3-min dissociation period with the same buffer, after which the sensor surface was regenerated by 40 μl of 2 M NaCl. The sensorgrams were fit with 1:1 Langmuir binding model from BIAevaluate 4.0.1.

**Structure-defined HS oligosaccharide microarray.** HS oligosaccharide array with 52 structure-defined HS oligosaccharides was prepared as previously

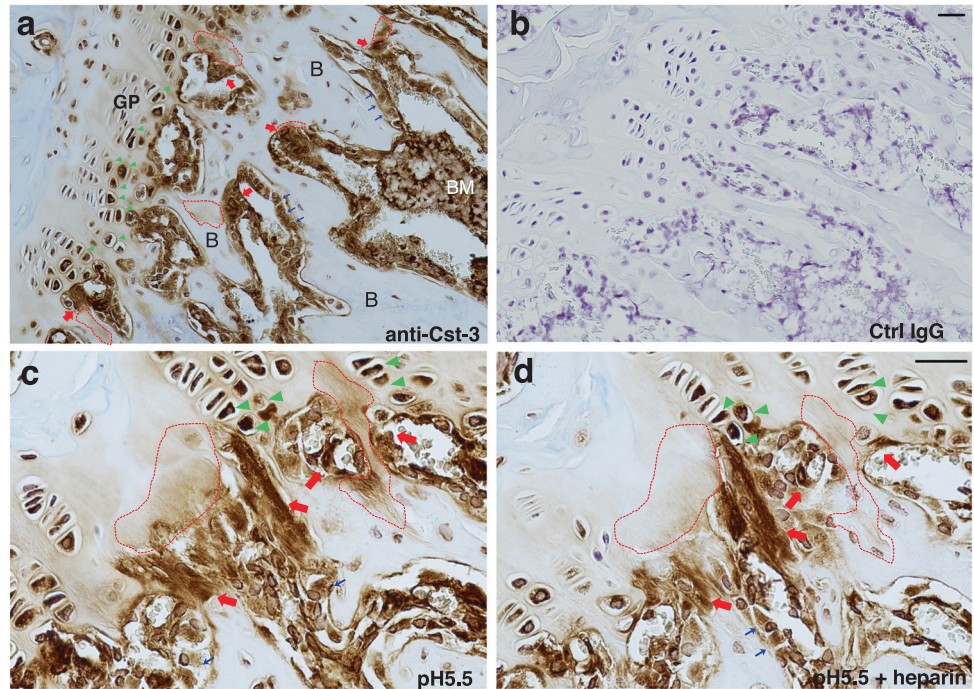

**Fig. 8 Cst-3 binds extracellular matrix HS in mouse bone tissue.** Mouse femur sections were immune-stained with either a goat anti-mouse Cst-3 polyclonal antibody (**a**) or control goat IgG (**b**). Osteoclasts are indicated with red arrows, chondrocytes with green triangles, and osteoblasts with blue arrows. B bone, BM bone marrow, GP grow plate. Magnification = ×200. **c, d** Two sequential bone sections were pretreated with pH 5.5 acetate buffer only (**c**) or together with heparin (**d**) prior to immunostaining with anti-mouse Cst-3. The red dotted lines encircle extracellular staining of Cst-3 that are sensitive to heparin treatment. Magnification = ×400. Scale = 15 μm.

described[45]. Briefly, the amine-containing HS oligosaccharides were printed onto NHS-activated glass slides. Each compound was printed at 6 × 6 pattern. The microarray was probed with 10 μg/ml Cst-3 in NaOAc buffer containing 150 mM NaCl at pH 5.5. Bound Cst-3 was stained with a biotinylated goat anti-mouse cystatin C (5 μg/ml, AF1238, R&D systems) followed by avidin conjugated with Alexafluor-488. The images were acquired using GenePix 4300 A scanner. The intensity data are the mean value ± S.D. of 36 individual spots. Of note, while heparin (compound #1) was intended to serve as positive control in the microarray, it showed poor binding to most HS-binding proteins that we tested[45–47]. This is likely due to the fact that heparin was immobilized through its internal free amines, which made it lay flat on the microarray surface. In contrast, all HS oligosaccharides were immobilized through its terminal amines, which gave them an upright configuration on the microarray surface.

**NMR spectrometry.** The backbone assignment of Cst-3 was obtained using standard triple resonance methods. Six 3D NMR experiments, HNCO, HN(CA) CO, HNCA, HN(CO)CA, HNCACB, and CBCA(CO)NH, were performed on a uniformly $^{13}$C and $^{15}$N-labeled Cst-3 at 0.3 mM, in 23 mM MES, 136 mM NaCl, 10% D2O, pH 6.6. All NMR experiments were performed on a Bruker 800 MHz spectrometer with a cryogenic probe. NMR data were processed using NMRPipe and analyzed with Sparky. Titration experiments with heparin and HS oligosaccharide were performed on $^{15}$N-labeled Cst-3. A series of s $^{1}$H–$^{15}$N-HSQC spectroscopy experiments were performed on a 0.3 mM Cst-3 sample by adding increasing amounts of heparin and HS oligosaccharide lyophilized aliquots.

**Site-directed mutagenesis.** Cst-3 mutants were prepared using published method[48]. Mutations were confirmed by sequencing, and recombinant protein was expressed as described for WT Cst-3. Purification was carried out using HiTrap SP cation exchange column at pH 6.5 (MES buffer), followed by gel permeation chromatography as described for WT Cst-3.

**Papain activity assays.** Papain (Sigma–Aldrich) was diluted to 130 nM in sodium acetate assay buffer (100 mM NaOAc, 2.5 mM EDTA, pH 5.5) and activated by addition of 5 mM DTT and room temperature incubation for 10 min. 4 μl fluorogenic substrate Z-Phe-Arg-pNA (Bachem, 10 mg/ml in DMSO) were mixed with 146 μl activated papain to initiate the reaction. WT or mutant Cst-3 were added concurrently with the substrate to reach a final concentration of 130 nM. A405 measurements were taken in a spectrophotometer (BioMate 3, Thermo Fisher) for 5 min with 1-min intervals. The enzymatic activity in the reaction was

calculated according to Lambert-Beer's Law. To measure the inhibitory activity of WT Cst-3 with or without HS 6mer/heparin, Cst-3 was used at a final concentration of 60 nM. In selected conditions, heparin or HS 6mer were preincubated with equal molar of Cst-3 at room temperature for 3 min before mixing with papain and the substrate.

When BSA was used as the substrate, BSA (Akron Biotech, 1 mg/ml in PBS) was denatured by heating at 80 °C for 10 min. In each sample, 5 μg denatured BSA were mixed with activated papain (final concentration 170 nM) to initiate the reaction. In selected samples, Cst-3 and/or HS 6mer were added to final concentration of 170 nM. All samples were incubated at 37 °C for 30 min to allow digestion. The reaction was stopped by heat denaturation at 98 °C for 10 min and the sample were resolved on a 4–20% Bis-Tris gel (Genscript).

**Immunohistochemistry.** Animal use was approved by the Institutional Animal Care and Use Committee (IACUC) of the University at Buffalo (Buffalo, NY). Mouse femurs from a 12.5-week-old male mouse were harvested and fixed for 48 h in 10% neutral buffered formalin, and decalcified in 10% EDTA for 2 weeks at room temperature. The samples were embedded in paraffin and sectioned at 5 μm. Each slide contained two serial tissue sections. After antigen retrieval and blocking step, sections were immuno-stained with either 3 μg/ml goat anti-mouse Cst-3 antibody (AF1238, R&D systems) or control goat IgG (Southern Biotech) overnight at 4 °C. For heparin competition assay, prior to antibody treatment, one section was incubated with 100 μM heparin (Sigma) in buffer containing 20 mM NaOAc, 150 mM NaCl, pH 5.5 and the other section was incubated with the same buffer without heparin overnight at 4 °C. After washing in PBS three times, they were treated for 1 h with biotinylated rabbit anti-goat IgG secondary antibody (1:200; Vector Laboratories). The sections were developed using the ABC system (Vector Laboratories) and the cell nuclei were counter stained with 15% Ehrlich's hematoxylin (Electron Microscopy Sciences).

**Statistics and producibility.** In Fig. 7, the data are expressed as means ± SD using four technical replicates. Statistical significance of differences between experimental groups was analyzed by two-tailed unpaired Student's $t$-test using GraphPad Prism software (GraphPad Software Inc.). $p$ value < 0.05 was considered significant. The flow cytometry experiments, the heparin–Sepharose chromatography and the histology straining were repeated at least three times with similar results. The representative data of these experiments were shown.

**Reporting summary**. Further information on research design is available in the Nature Research Reporting Summary linked to this article.

## Data availability
The source data for generating Figs. 3b and 7a are included as Supplemental Data 1 and 2, respectively. The Sequencing data for Cst-3 mutants can be assessed at GenBank MW488011-MW488017. Any remaining information can be obtained from the corresponding author upon reasonable request.

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

## Acknowledgements
The work is supported by NIH grants AR070179 to D.X., HL094463 to D.X. and J.L., and AG069039 to C.W. The content is solely the responsibility of the authors and does not necessarily represent the official views of the National Institutes of Health.

## Author contributions
Z.X., X.L., J.L., C.W., and D.X. designed the research, Z.X., X.L., G.S., and M.L. performed the work, Z.X., X.L., G.S., C.W., and D.X. analyzed and interpreted data, Z.X., X.L., C.W., and D.X. wrote the manuscript. J.L. revised the paper.

## Competing interests

The authors declare no competing interests.
