## [Peer Review File · Communications Biology]

Reviewers' comments:

Reviewer #1 (Remarks to the Author):

In their manuscript, Zhang et al. identify that Cystatin C as a new heparan sulfate (HS) binding protein. Initially, the authors used heparin Sepharose affinity chromatography as a proxy to determine Cystatin C binding but only at low pH, and confirmed HS binding by FACS analysis using HS-expressing CHO-K1 cells. SPR analyses determined that Cystatin C binding to HS/heparin strongly depends on pH, and an HS oligosaccharide microarray was also used to better determine the prerequisites of Cystatin C/HS interactions. This revealed that HS oligosaccharides as small as hexamers were already bound and that overall sulfation did not appear to determine binding. Finally, by NMR and mutagenesis analysis, the authors identified 2 isolated HS binding sites on the protein that require to come together to achieve significant HS binding. This is a property of HS binding proteins not previously described for the plethora of known HS binding proteins.

The authors present a paper that is of high technical quality and also of high interest to the field, because Cystatin C behaves different from other heparan sulfate binding proteins in several aspects, most notably in having two HS binding sites that are both required for binding, and the pH dependency of the interaction. The manuscript is also very well written and is very logical in style. I do have one major and some minor comments regarding the findings that address some of the shortcomings. If comments are addressed, this would definitely strengthen their results and conclusions.

Major comment:

The authors describe pH dependency of Cystatin/HS interactions, but leave the question open what relevance this interactions might have on the activity of the protein. To this reviewer, the absence of Cystatin/HS interactions in most physiological settings indicates that this interaction might interfere with its inhibitory activity on cysteine proteases. At low pH, therefore, Cystatin binding to HS may increase proteolytic activities in this area. I believe the effort to test this question (in vitro, using available cysteine protease activity tests) would reveal mechanistical insight that would strengthen this story a lot.

Minor comments:

CHO cells were used to determine Cystatin/HS interactions and binding was inhibited with heparin as a control. The authors are certainly aware of the various HS-deficient CHO cell lines that are available - have these mutant lines been used to confirm that HS is the relevant glycosaminoglycan bound?

Figure 2: Labelling of the SPR sensorgrams may be improved to make it easier on the reader to identify the conditions used in the experiment (the lowest concentrations should move to the bottom). Also, the units should be the same (0.625 micromolar in one figure versus 625 nanomolar). Maybe I am mistaken, but figure legend 2 says "...the highest concentration of...is half of that..." Should it not read "the lowest"? I am not an expert, but the 20micromolar values (highest) look pretty much the same to me, the lowest (312 and 625 nanomolar) differ by half.

Does the large binding site identified by NMR/mutagenesis match the HS hexamer size identified on the HS chip?

I hope these comments help to improve the manuscript. I am looking forward to the authors reply.

Reviewer #2 (Remarks to the Author):

This communication describes for the first time the interaction between cystatin C (Cst-3) and heparin/heparan sulfate (HS). Affinity chromatography and surface plasmon resonance demonstrate an interaction with modest (K_d 0.6 micromolar) affinity at pH 5.5, and with reduced affinity at pH 6.5. Flow cytometry indicated binding of Cst-3 to CHO-K1 cells at pH 6.5 but not at pH 7.1. A microarray analysis of Cst-3 binding to 52 HS-like oligosaccharides identified a heptasulfated hexasaccharide as a ligand, and this oligosaccharide was then used in NMR titration experiments to determine which amino acid residues in Cst-3 are involved in heparin/HS binding. Residues in the unstructured N-terminal sequence, and a loop region nearer the C-terminus, were identified and mutagenesis experiments carried out to assess the importance of these residues to the HS-Cst interaction. Mutant proteins with reduced binding to heparin also had reduced capacity to interact with CHO-K1 cells. Treatment of mouse femur sections with heparin at pH 5.5 reduced Cst-3 staining in bone matrix while having no effect on Cst-3 staining in osteoclasts, chondrocytes and osteoblasts.

Altogether a reasonable case is made for the presence of a heparin/HS-binding site on the surface of Cst-3 made up of at least two clusters of basic residues, one cluster in an unstructured tail and one in a semi-structured loop and helix region near His-90. The pH dependence of binding is demonstrated and attributed, again reasonably, to a histidine switch within one of the basic clusters. The potential significance of the interaction in biology is hinted at by the disruption of Cst-3 binding to bone matrix in the presence of heparin at low pH, and followed up by the authors with a speculative discussion. Both Cst-3 and HS take part in numerous physiological processes, many of which involve acidic environments.

There are some questions that arise that should be addressed in a revised manuscript.

1. The microarray experiments: What do the error bars in Fig. 3B signify? The caption and the Methods section do not make this clear. Are the differences between hexasaccharide 34 and, for example, 23 or 41 quantitatively significant? It is disappointing that the array does not do more to aid identification of a structural motif in HS that conveys affinity for Cst-3. Position 1 in the microarray is labelled 'heparin' but binding to it seems very low; can this be explained? This procedure seems like a rather elaborate method for finding a suitable heparin fragment for use in NMR titration.
2. The hexasaccharide is quite a small structure with respect to the full extent of the proposed binding site, going by measurements of the NMR structure 6RPV. Even when the N-terminal tail points directly towards the H90 basic cluster, the distance between His90 and Arg8 is over 3 nm, rather longer than the expected length of the hexasaccharide. Was the affinity of the hexasaccharide for Cst-3 measured? A longer oligosaccharide might have been better for NMR studies. The shifts observed on heparin titration are rather small. See for comparison the study of Sepuru et al. *J Biol Chem.* 2018 Nov 16;293(46):17817-17828.
3. In the NMR-derived models that place the N-terminus near to the H90 cluster (6rpv models 2, 12, 16), the protein surface displays a promising linear arrangement of surface positive charge extending from K8 to H90, and including some residues not investigated in this study such as K92 and R51. Is there a reason that these residues were not included in the mutagenesis study?
4. The authors suggest that heparin might induce helical structure in the N-terminal tail. In the NMR models referred to above there is no helical aspect to the N-terminal tail. If the hexasaccharide induces this, the ability of the tail to reach towards the H90 cluster will be reduced and the two halves of the binding site may not meet up (though of course, full length heparin would easily bridge the gap).
5. Mutations can affect heparin binding either because the altered residues are directly involved in the interaction or because they are important for the conformation of the protein. Is there evidence that the Cst-3 mutants retain their structural integrity?

6. In the Methods section, the microarray section is very brief and sends the reader to reference numbers 42 and 43 that are not relevant. This section should be corrected and extended to contain sufficient detail so that the reader can make sense of Fig. 3 (for example the meaning of the error bars). Also, is reference 44 correct for site-directed mutagenesis?

Reviewer #3 (Remarks to the Author):

Dear Authors,

The publication submitted to me for review concerns the study of the interaction of cystatin C with Heparan Sulfate (HS). This is a new topic because no one has studied the interaction of cystatin C with HS so far. The results of this work could play a role in understanding how this protein works at the tissue level. The publication uses a wide variety of research techniques, from protein overproduction, through NMR spectroscopy, molecular filtration techniques, and in vivo testing and staining. I have some comments and remarks to the publication presented for evaluation:

Major points:

1) The title of the publication and the introduction are written in such a way that the reader is convinced that the research concerns human Cystatin C. Only when we go to the reading of the results it turns out that the research concerns mouse cystatin C, which has a homology level of about 70% to human cystatin C (hCC). In my opinion, there is a lack of orderliness and directing the reader to the fact that mouse cystatin C is being tested. It would be appropriate to write in the introduction to the publication what are the differences between the two proteins and why the mouse protein is tested. While reading the publication, it turned out that the authors made an attempt to overproduce human cystatin C in human cells, but without much success. Mouse cystatin C was produced in E. coli bacterial cells. Why has no attempt been made to overproduce human cystatin C in E.coli. It is known that many scientists receive human Cystatin C in this way.

2) In my opinion, the introduction to the work does not contain the research hypothesis. The authors investigate the interaction of cystatin C with heparan sulfate (HS), but we don't know why? This information is only clarified in the conclusions but it should not be. It might as well study any other impact. First, the reader should find out why such a topic was taken up in the reviewed publication and why the influence of different pH on the degree of protein binding to HS was studied.

3) In the introduction (page 2) there is information that Cystatin C has a pI = 8. But it is not known which Cystatin C is affected by this value?

4) Figure 1 and the text relating to it shows two different proteins tested by two different techniques. The reason is the too little amount of human cystatin C that the authors received. In my opinion, there is no point in showing the results for human cystatin C, since further research only looks at the murine version of this protein. In my opinion, Western blots should be performed and reported for murine cystatin C.

5) How do the authors explain the fact that the His86Ala mutant behaves like a native protein and the His90Ala mutant already has a much lower impact? Both His are very close to each other and their protonation pH is very similar. In my opinion, we should also discuss the protein structure in this region and see if the His86 residue is exposed outside the protein and if it has a "chance" to interact with HS.

6) There is also a discussion in the publication about the distance between the N-terminus of the

protein and His90 and that the N-terminus somehow necessitates an interaction with the HS of the His90 residue. Since authors know how long the HS chains must be for cystatin C to bind, it may be worth checking whether the stretched HS chain can interact with both parts of the protein simultaneously.

7) Figure 4. Why is the enlarged region of the NMR spectrum for His90 missing?

Minor points:

8) Figure 3. The letters in the figure are too small to read them. They should be enlarged.

9) Figure 4. In the description of the spectrum (panel A) there is Cys C-oligo. The word "oligo" suggests the formation of oligomers by Cystatin C. It is known that this is the natural behavior of human cystatin C and therefore this panel should be described differently so that it does not suggest oligomeric forms of the protein.

Reviewer #1 (Remarks to the Author):

In their manuscript, Zhang et al. identify that Cystatin C as a new heparan sulfate (HS) binding protein. Initially, the authors used heparin Sepharose affinity chromatography as a proxy to determine Cystatin C binding but only at low pH, and confirmed HS binding by FACS analysis using HS-expressing CHO-K1 cells. SPR analyses determined that Cystatin C binding to HS/heparin strongly depends on pH, and an HS oligosaccharide microarray was also used to better determine the prerequisites of Cystatin C/HS interactions. This revealed that HS oligosaccharides as small as hexamers were already bound and that overall sulfation did not appear to determine binding. Finally, by NMR and mutagenesis analysis, the authors identified 2 isolated HS binding sites on the protein that require to come together to achieve significant HS binding. This is a property of HS binding proteins not previously described for the plethora of known HS binding proteins.

The authors present a paper that is of high technical quality and also of high interest to the field, because Cystatin C behaves different from other heparan sulfate binding proteins in several aspects, most notably in having two HS binding sites that are both required for binding, and the pH dependency of the interaction. The manuscript is also very well written and is very logical in style. I do have one major and some minor comments regarding the findings that address some of the shortcomings. If comments are addressed, this would definitely strengthen their results and conclusions.

Major comment:

The authors describe pH dependency of Cystatin/HS interactions, but leave the question open what relevance this interactions might have on the activity of the protein. To this reviewer, the absence of Cystatin/HS interactions in most physiological settings indicates that this interaction might interfere with its inhibitory activity on cysteine proteases. At low pH, therefore, Cystatin binding to HS may increase proteolytic activities in this area. I believe the effort to test this question (in vitro, using available cysteine protease activity tests) would reveal mechanistical insight that would strengthen this story a lot.

We thank the reviewer for this excellent suggestion. We have tested the inhibitory capability of Cst-3 towards papain in the presence or absence of HS and found that the HS-bound Cst-3 was unable to inhibit papain anymore (new Fig. 7), just as the reviewers suspected. This result can be explained by the dual roles of the N-terminal peptide in mediating both protease inhibition and HS-binding. The role of N-terminus in protease inhibition is demonstrated in Fig. S4, where we showed that the N-terminal deletion mutant greatly reduces its efficacy as a cysteine protease inhibitor. It is likely HS binds the flexible N-terminus and causes a large conformational change, likely weakening Cst-3/papain interaction and the efficacy of protease inhibition.

Figure 7. HS interferes with the biological function of Cst-3. (A) The enzymatic activity of papain was determined by using a colorimetric peptide substrate in the presence or absence of Cst-3, and in the presence or absence of HS hexasaccharide (6mer) and heparin (Hp). $n = 4$. (B) Heat-denatured BSA was digested with papain in the presence or absence of Cst-3, and/or in the presence or absence of HS 6mer. N.Ctrl lane was run with only denatured BSA. Data is representative of at least three similar experiments

Minor comments:

CHO cells were used to determine Cystatin/HS interactions and binding was inhibited with heparin as a control. The authors are certainly aware of the various HS-deficient CHO cell lines that are available - have these mutant lines been used to confirm that HS is the relevant glycosaminoglycan bound?

We have now performed Cst-3 binding to pgsE and pgsF cells and this new result was presented in Fig. 1E. Briefly, we found that binding Cst-3 to pgsE was reduced by 70% compared to its binding to CHO-K1 cells. In pgsE cell line, sulfation levels at all positions are greatly decreased; therefore this suggest that Cst-3 binding is highly sensitive to the overall sulfation level of HS on CHO cell surface. In contrast, the binding of Cst-3 to pgsF cells was comparable to its binding to CHO-K1 cells. As pgsF cells specifically lack 2-O-sulfation but maintains a similar level of overall sulfation as CHO-K1 cells, this result suggests that 2-O-sulfation is not required for Cst-3/HS interaction, which is consistent with our microarray analysis.

Fig. 1(E) Binding of Cst-3 to pgsE and pgsF, two mutant CHO cell lines with altered HS structures.

Figure 2: Labelling of the SPR sensorgrams may be improved to make it easier on the reader to identify the conditions used in the experiment (the lowest concentrations should move to the bottom). Also, the units should be the same (0.625 micromolar in one figure versus 625 nanomolar). Maybe I am mistaken, but figure legend 2 says "...the highest concentration of...is half of that..." Should it not read "the lowest"? I am not an expert, but the 20micromolar values (highest) look pretty much the same to me, the lowest (312 and 625 nanomolar) differ by half.

We thank the reviewer for this suggestion. The figure and legend have been modified accordingly.

Does the large binding site identified by NMR/mutagenesis match the HS hexamer size identified on the HS chip?

A typical HS hexasaccharide measures $\sim 27 \text{ \AA}$. This size is similar to that of HS binding site in Cst-3, which measures $\sim 30 \text{ \AA}$ in length. We also expect a conformational change in Cst-3 upon HS binding, especially at and near the N-terminus, which could further shorten the HS-binding site in Cst-3 to fit the hexasaccharide.

I hope these comments help to improve the manuscript. I am looking forward to the authors reply.

Reviewer #2 (Remarks to the Author):

This communication describes for the first time the interaction between cystatin C (Cst-3) and heparin/heparan sulfate (HS). Affinity chromatography and surface plasmon resonance demonstrate an interaction with modest (K_d 0.6 micromolar) affinity at pH 5.5, and with reduced affinity at pH 6.5. Flow cytometry indicated binding of Cst-3 to CHO-K1 cells at pH 6.5 but not at pH 7.1. A microarray analysis of Cst-3 binding to 52 HS-like oligosaccharides identified a heptasulfated hexasaccharide as a ligand, and this oligosaccharide was then used in NMR titration experiments to determine which amino acid residues in Cst-3 are involved in heparin/HS binding. Residues in the unstructured N-terminal sequence, and a loop region nearer the C-terminus, were identified and mutagenesis experiments carried out to assess the importance of these residues to the HS-Cst interaction. Mutant proteins with reduced binding to heparin also had reduced capacity to interact with CHO-K1 cells. Treatment of mouse femur sections with heparin at pH 5.5 reduced Cst-3 staining in bone matrix while having no effect on Cst-3 staining in osteoclasts, chondrocytes and osteoblasts. Altogether a reasonable case is made for the presence of a heparin/HS-binding site on the surface of Cst-3 made up of at least two clusters of basic residues, one cluster in an unstructured tail and one in a semi-structured loop and helix region near His-90. The pH dependence of binding is demonstrated and attributed, again reasonably, to a histidine switch within one of the basic clusters. The potential significance of the interaction in biology is hinted at by the disruption of Cst-3 binding to bone matrix in the presence of heparin at low pH, and followed up by the authors with a speculative discussion. Both Cst-3 and HS take part in numerous physiological processes, many of which involve acidic environments. There are some questions that arise that should be addressed in a revised manuscript.

1. The microarray experiments: What do the error bars in Fig. 3B signify? The caption and the Methods section do not make this clear. Are the differences between hexasaccharide 34 and, for example, 23 or 41 quantitatively significant? It is disappointing that the array does not do more to aid identification of a structural motif in HS that conveys affinity for Cst-3. Position 1 in the microarray is labelled 'heparin' but binding to it seems very low; can this be explained? This procedure seems like a rather elaborate method for finding a suitable heparin fragment for use in NMR titration.

The error bar indicates variations in 36 replicate spots of oligosaccharides printed on the microarray. We have added this information in both Legend and Method sections.

We chiefly use the oligosaccharide microarray as a qualitative method to identify the structures that are capable of binding certain HS-binding protein. Comparing the bars in a quantitative way might lead to overinterpretation of the binding specificity. In this case, we believe oligo #34, 23 and 41 can all bind Cst-3 reasonably well and using any of them for NMR titration would be appropriate.

With regards to the heparin spot, the low binding to Cst-3 was most likely due to that heparin was immobilized by using internal free amines, whereas the oligosaccharides were immobilized by using a terminal free amine. As a result, the immobilized heparin lies flat on the chip surface, which interferes with its accessibility for interaction; while the oligosaccharides stand vertically on the chip. Based on our experience, most HS-binding proteins that we tested on the microarray bound poorly to the immobilized heparin, including antithrombin, S100A12 and Tau.

2. The hexasaccharide is quite a small structure with respect to the full extent of the proposed binding site, going by measurements of the NMR structure 6RPV. Even when the N-terminal tail points directly towards the H90 basic cluster, the distance between His90 and Arg8 is over 3 nm, rather longer than the expected length of the hexasaccharide. Was the affinity of the hexasaccharide for Cst-3 measured? A longer oligosaccharide might have been better for NMR studies. The shifts observed on heparin titration are rather small. See for comparison the study of Sepuru et al. *J Biol Chem.* 2018 Nov 16;293(46):17817-17828.

The length of the hexasaccharide is ~ 2.7 nm, which would nicely occupy the proposed HS-binding site with a length of around 3 nm as the reviewer pointed out. Also, due to the dynamic nature of the H90 cluster and the N-terminal tail, it's difficult to predict the exact distance between H90 and R8, which could be closer than what's depicted in figure 6. We also expect a conformational change in Cst-3 upon HS binding, especially at and near the N-terminus, which may further shorten the binding site to fit the hexasaccharide. Given that the binding signal of Cst-3 to hexasaccharides #34 are among the best in our microarray analysis, we felt it's unlikely that longer oligosaccharide would provide benefit for our NMR studies.

The chemical shifts observed in heparin titration, presented in SI2, are significant. E.g., some N-terminal CSPs are bigger than 1 ppm, while CSPs at other heparin binding sites reach ~0.2 to 0.4 ppm. Thus these shifts are comparable, or even bigger, than those presented in Sepuru et al. Hexasaccharide titration led to smaller CSP but still reaches ~0.3 ppm in the ¹⁵N dimension in the N-terminus.

3. In the NMR-derived models that place the N-terminus near to the H90 cluster (6rpv models 2, 12, 16), the protein surface displays a promising linear arrangement of surface positive charge extending from K8 to H90, and including some residues not investigated in this study such as K92 and R51. Is there a reason that these residues were not included in the mutagenesis study?

In murine Cst-3, the residue at 92 is actually a Met. K92 is only present in human Cst-3. We actually performed R51A mutant and found it is also involved in the binding. This information was added into Table II and Fig. 6A.

4. The authors suggest that heparin might induce helical structure in the N-terminal tail. In the NMR models referred to above there is no helical aspect to the N-terminal tail. If the hexasaccharide induces this, the ability of the tail to reach towards the H90 cluster will be reduced and the two halves of the binding site may not meet up (though of course, full length heparin would easily bridge the gap).

We thank the reviewer for raising a good point, although this is a minor point in the discussion. A solid conclusion about the helical content needs to be further validated by other experimental techniques such as CD or structure determination of the complex. Helical formation will indeed compress the HS-binding sites in the N-terminus, making them further away if there is no further conformational change associated with HS binding. However, additional conformational change is quite likely, which may bring to HS binding sites to the right place. In the revised MS, we changed the wording "helical content" to "helical propensity".

5. Mutations can affect heparin binding either because the altered residues are directly involved in the interaction or because they are important for the conformation of the protein. Is there evidence that the Cst-3 mutants retain their structural integrity?

We have tested the activity of cst-3 towards papain and found that except the N-terminal deletion mutant, the activities of all other mutants were intact, validating their structure integrity. This new information is now included as Supplemental Figure 4.

6. In the Methods section, the microarray section is very brief and sends the reader to reference numbers 42 and 43 that are not relevant. This section should be corrected and extended to contain sufficient detail so that the reader can make sense of Fig. 3 (for example the meaning of the error bars). Also, is reference 44 correct for site-directed mutagenesis?

We thank the reviewer for this suggestion. This section has now been extended to contain more experimental details of the microarray analysis.

Reviewer #3 (Remarks to the Author):

Dear Authors,

The publication submitted to me for review concerns the study of the interaction of cystatin C with Heparan Sulfate (HS). This is a new topic because no one has studied the interaction of cystatin C with HS so far. The results of this work could play a role in understanding how this protein works at the tissue level. The publication uses a wide variety of research techniques, from protein overproduction, through NMR spectroscopy, molecular filtration techniques, and in vivo testing and staining. I have some comments and remarks to the publication presented for evaluation:

Major points:

1) The title of the publication and the introduction are written in such a way that the reader is convinced that the research concerns human Cystatin C. Only when we go to the reading of the results it turns out that the research concerns mouse cystatin C, which has a homology level of about 70% to human cystatin C (hCC). In my opinion, there is a lack of orderliness and directing the reader to the fact that mouse cystatin C is being tested. It would be appropriate to write in the introduction to the publication what are the differences between the two proteins and why the mouse protein is tested. While reading the publication, it turned out that the authors made an attempt to overproduce human cystatin C in human cells, but without much success. Mouse cystatin C was produced in *E. coli* bacterial cells. Why has no attempt been made to overproduce human cystatin C in *E. coli*. It is known that many scientists receive human Cystatin C in this way.

We thank the reviewer for this suggestion and has now revised the Introduction section to discuss the homology between human and murine Cst-3 (Supplemental Fig. 3) and why murine Cst-3 was used in the study.

We have actually produced human Cst-3 in *E. coli*. However all purified recombinant human Cst-3 existed in the dimeric form. In contrast, the majority of *E. coli* expressed murine Cst-3 exists in monomeric form. As the dimeric form of Cst-3 is known to be inactive as a protease inhibitor, we chose to study the monomeric form of murine Cst-3 instead. Of note, we have confirmed that the dimeric human Cst-3 binds heparin Sepharose with a higher affinity than the dimeric murine Cst-3, which were eluted by 560 mM and 410 mM NaCl at pH5.5, respectively (Figure for the reviewers).

Figure for reviewers: *E. coli* expressed human and murine Cst-3 dimer were purified by Superdex200 and the dimer fractions were run on HiTrap heparin Sepharose at pH5.5. Human Cst-3 dimer displayed significantly higher binding (eluted by 560 mM NaCl) to heparin Sepharose compared to murine Cst-3 dimer (eluted by 410 mM NaCl).

2) In my opinion, the introduction to the work does not contain the research hypothesis. The authors

investigate the interaction of cystatin C with heparan sulfate (HS), but we don't know why? This information is only clarified in the conclusions but it should not be. It might as well study any other impact. First, the reader should find out why such a topic was taken up in the reviewed publication and why the influence of different pH on the degree of protein binding to HS was studied.

We thank the reviewer for this suggestion. We have revised the introduction to make our hypothesis more clear.

3) In the introduction (page 2) there is information that Cystatin C has a pI = 8. But it is not known which Cystatin C is affected by this value?

Murine Cst-3 has a predicted pI = 8.75, while human Cst-3 has a predicted pI = 8.5.

4) Figure 1 and the text relating to it shows two different proteins tested by two different techniques. The reason is the too little amount of human cystatin C that the authors received. In my opinion, there is no point in showing the results for human cystatin C, since further research only looks at the murine version of this protein. In my opinion, Western blots should be performed and reported for murine cystatin C.

We apologize for this confusion. The WB shown was actually using murine Cst-3 expressed in 293 cells.

5) How do the authors explain the fact that the His86Ala mutant behaves like a native protein and the His90Ala mutant already has a much lower impact? Both His are very close to each other and their protonation pH is very similar. In my opinion, we should also discuss the protein structure in this region and see if the His86 residue is exposed outside the protein and if it has a "chance" to interact with HS.

We thank the reviewer for this suggestion. We believe the most likely explanation would be that H86 is located far from the identified HS binding site. When HS binds to the identified HS-binding site, HS is unlikely to with H86 because H86 is ~ 6 Å from the H90.

6) There is also a discussion in the publication about the distance between the N-terminus of the protein and His90 and that the N-terminus somehow necessitates an interaction with the HS of the His90 residue. Since authors know how long the HS chains must be for cystatin C to bind, it may be worth checking whether the stretched HS chain can interact with both parts of the protein simultaneously.

A typical HS hexasaccharide measures ~27 Å. This size is similar to that of HS binding site in Cst-3, which measures ~30 Å in length. We also expect a conformational change in Cst-3 upon HS binding, especially at and near the N-terminus, which can further shorten the HS-binding site in Cst-3 to fit the hexasaccharride.

7) Figure 4. Why is the enlarged region of the NMR spectrum for His90 missing?

We thank the reviewer for pointing this out. We actually didn't have the NMR assignment for H90. The involvement of H90 in heparin or HS binding was deduced from CSP in nearby residues, L91, R93 and K94, labeled in panel A, Fig. 4. We also replotted panel B and C to clearly show that we don't have CSP data for a few residues (shown as without data bar), including H90.

We mislabeled R93 in the original submission as N35, which has now been corrected in the revised figure 4.

Minor points:

8) Figure 3. The letters in the figure are too small to read them. They should be enlarged.

The font has been enlarged.

9) Figure 4. In the description of the spectrum (panel A) there is Cys C-oligo. The word "oligo" suggests the formation of oligomers by Cystatin C. It is known that this is the natural behavior of human cystatin C and therefore this panel should be described differently so that it does not suggest oligomeric forms of the protein.

We thank the reviewer for this suggestion and we have changed the label to Cst-3:HS.

REVIEWERS' COMMENTS:

Reviewer #1 (Remarks to the Author):

The authors have adequately addressed all of my concerns.

Reviewer #2 (Remarks to the Author):

Thanks to the authors for the clarifications and additions they have made to this manuscript. All the minor points raised by this referee have been dealt with. Due to the suggestion of another referee, the message of the paper is now substantially more significant, as HS is now shown not only to bind to Cst-3 but to impede its protease inhibitory activity at low pH. The combination of structural and functional biology makes a satisfactory story, well presented and concisely written.

Reviewer #3 (Remarks to the Author):

Dear Authors,

I noticed that the publication was greatly improved by the authors. I have no more comments.